# Eating Speed Is Associated with the Presence of Sarcopenia in Older Patients with Type 2 Diabetes: A Cross-Sectional Study of the KAMOGAWA-DM Cohort

**DOI:** 10.3390/nu14040759

**Published:** 2022-02-11

**Authors:** Yoshitaka Hashimoto, Fuyuko Takahashi, Ayumi Kaji, Ryosuke Sakai, Takuro Okamura, Noriyuki Kitagawa, Hiroshi Okada, Naoko Nakanishi, Saori Majima, Takafumi Senmaru, Emi Ushigome, Mai Asano, Masahide Hamaguchi, Masahiro Yamazaki, Michiaki Fukui

**Affiliations:** 1Department of Endocrinology and Metabolism, Graduate School of Medical Science, Kyoto Prefectural University of Medicine, 465, Kajii-cho, Kawaramachi-Hirokoji, Kamigyo-ku, Kyoto 602-8566, Japan; fuyuko-t@koto.kpu-m.ac.jp (F.T.); kaji-a@koto.kpu-m.ac.jp (A.K.); sakaryo@koto.kpu-m.ac.jp (R.S.); d04sm012@koto.kpu-m.ac.jp (T.O.); nori-kgw@koto.kpu-m.ac.jp (N.K.); conti@koto.kpu-m.ac.jp (H.O.); naoko-n@koto.kpu-m.ac.jp (N.N.); saori-m@koto.kpu-m.ac.jp (S.M.); semmarut@koto.kpu-m.ac.jp (T.S.); emis@koto.kpu-m.ac.jp (E.U.); maias@koto.kpu-m.ac.jp (M.A.); mhama@koto.kpu-m.ac.jp (M.H.); masahiro@koto.kpu-m.ac.jp (M.Y.); michiaki@koto.kpu-m.ac.jp (M.F.); 2Department of Diabetology, Kameoka Municipal Hospital, 1-1 Noda, Shinochoshino, Kyoto 621-8585, Japan; 3Department of Diabetes and Endocrinology, Matsushita Memorial Hospital, 5-55 Sotojima-cho, Moriguchi 570-8540, Japan

**Keywords:** eating speed, diet, muscle mass, sarcopenia, diabetes

## Abstract

To determine the relationship between eating speed and the presence of sarcopenia in older patients with type 2 diabetes (T2D), in this cross-sectional study, patient eating speeds were classified as “fast-”, “normal-” and “slow-speed eating.” A multifrequency impedance analyzer was used to evaluate patient body compositions. Sarcopenia was defined as having both low muscle strength, a handgrip strength <28 kg for men and <18 kg for women, and low skeletal muscle mass as a skeletal muscle mass index <7.0 kg/m^2^ for men and <5.7 kg/m^2^ for women. Among 239 individuals, the frequencies of fast-, normal-, and slow-speed eating were 47.3%, 32.2%, and 20.5%, respectively; and the prevalence of sarcopenia was 15.9%. Patients with a slow eating speed had greater prevalence of low skeletal muscle mass, low muscle strength, and sarcopenia than those with a fast or normal eating speed. After adjusting for covariates, compared to slow eaters, the odds ratio of having sarcopenia among fast- and normal-speed eaters was 0.31 [95% CI: 0.12–0.80] and 0.18 [95% CI: 0.06–0.53], respectively. Having a slow eating speed is associated with a heightened risk of sarcopenia in older patients with T2D.

## 1. Introduction

The number of older patients with type 2 diabetes (T2D) has grown in recent years [1,2] and is forecasted to accelerate over the next few decades [3], owing to people living longer [4] and an increase in the prevalence of T2D within the general population [5]. A plethora of microvascular and macrovascular complications have been documented to emerge following the diagnosis of T2D in older individuals, one of the newest being sarcopenia [2,6], an age-related degeneration of skeletal muscle that raises the likelihood of early mortality [7,8].

Heeding this, the identification of risk factors for sarcopenia is critical for improving future prognoses. Two meta-analyses published last year revealed that among patients with T2D, increasing age, being male, or having hyperglycemia, osteoporosis, excessive visceral fat mass, diabetic nephropathy, T2D for a long duration, or high levels of C-reactive protein was associated significantly with an elevated risk of developing sarcopenia [9,10]. The contribution of one factor that remains to be fully clarified, however, is that of a person’s dietary behavior.

Concerning consumption patterns, our team and several others have reported previously that low intakes of energy, proteins, vitamins B1, B12, and D, and omega-3 fatty acids augmented the chances of sarcopenia in older patients with T2D [11,12,13,14,15,16]. Eating speed, on the other hand, has attracted little attention to our knowledge, to date, even though there is compelling evidence to suggest that it exerts a prominent influence on the pathogenesis of the aforementioned disease. In particular, research shows that eating fast is connected to high energy intake [17] and a high body weight [18], which in turn may help attenuate the wasting effects of sarcopenia. For this cross-sectional investigation, we evaluated the relationship between eating speed and the presence of sarcopenia in older patients with T2D, hypothesizing that eating speed would be inversely associated with the presence of sarcopenia.

## 2. Materials and Methods

### 2.1. Study Participants

Individuals in this study were outpatients of Kyoto Prefectural University of Medicine Hospital (Kyoto, Japan) or Kameoka Municipal Hospital (Kameoka, Japan) and part of the KAMOGAWA-DM cohort, who have received ongoing assessments since 2014 by our group to yield insights into the history and progression of T2D [19]. Eligibility criteria consisted of having T2D and providing written informed consent to participate in the study; exclusion criteria consisted of not having T2D, using steroids, being under 60 years of age, or having medical charts that lacked data on handgrip strength, body composition, and/or habitual dietary intakes. The study was approved by the ethics committee of Kyoto Prefectural University of Medicine (No. RBMR-E-466-5) and was conducted in accordance with the Declaration of Helsinki.

### 2.2. Data Collection

A standardized questionnaire was used to gather data on each patient’s duration of T2D, and smoking and exercise habits (i.e., frequency of playing sports on a weekly basis). (See Appendix A) According to this information, patients were classified as non- or current smokers, and non- or regular venous blood draws took place after the patients had fasted overnight to ascertain their fasting plasma glucose and hemoglobin A1c levels. The estimated glomerular filtration rate (eGFR; mL/min/1.73 m^2^) was calculated by the Japanese Society of Nephrology equation [20] and eGFR under 30 mL/min/1.73 m^2^ were defined as chronic kidney diseases (CKD) stage ≥4 [21]. Past medical histories regarding heart diseases, including angina, coronary heart disease, heart failure, prior acute myocardial infarction, stroke (ischemic or hemorrhagic), and cancers, were obtained from electronic medical records. Data on medications for T2D (e.g., insulin, glucagon-like peptide-1, antagonist sodium-glucose cotransporter-2 inhibitors, and steroids) were also obtained from medical records.

### 2.3. Assessments of Eating Speed and Habitual Dietary Intakes

The brief-type self-administered diet history questionnaire [22] was employed to assess patient eating speed and habitual dietary intakes of 58 foods and beverages over the past month. Eating speed was evaluated by the following statements: for the former, the descriptors “very fast”, “a little fast”, “normal”, “a little slow”, or “very slow” were selected by the participants to characterize their eating speed, and then this information was used to classify the participants as “fast-speed eating”, “normal-speed eating”, or “slow-speed eating” [23]. Regarding the last, we obtained data on each participant’s nutrient intake and then calculated their total energy intake by dividing the total energy intake by ideal body weight (kcal/IBW/day) [23]. Carbohydrate (% energy), protein (% energy), and fat (% energy) intakes were calculated by multiplying the carbohydrate intake (g/day) by 4 (kcal/g) and dividing that by the total energy intake (kcal/day), multiplying the protein intake (g/day) by 4 (kcal/g) and dividing that by the total energy intake (kcal/day), and multiplying the fat intake (g/day) by 9 (kcal/g) and dividing that by the total energy intake (kcal/day), respectively.

### 2.4. Assessment of Sarcopenia

InBody 720 (InBody Japan, Tokyo, Japan), a multifrequency impedance body composition analyzer with accuracy comparable to dual-energy X-ray absorptiometry [24], was used to evaluate body composition. Briefly, the body weight (kg) and appendicular muscle mass (kg) were measured by this analyzer, and then the body mass index (BMI, kg/m^2^) or skeletal muscle mass index (kg/m^2^) was calculated by dividing the body weight (kg) or appendicular muscle mass (kg), respectively, by height squared (m^2^).

A handgrip dynamometer (Smedley, Takei Scientific Instruments Co., Ltd., Niigata, Japan) evaluated patient handgrip strength, and the maximum value of handgrip strength for both hands was used in our analysis.

Low muscle strength was defined as having a handgrip strength <28 kg for men and <18 kg for women; low skeletal muscle mass was defined as having a skeletal muscle mass index <7.0 kg/m^2^ for men and <5.7 kg/m^2^ for women; sarcopenia was defined as having both low muscle strength and low skeletal muscle mass [25].

### 2.5. Statistical Analyses

We used JMP 13.2 software (SAS, Cary, NC, USA) for statistical analyses and considered *p* < 0.05 as statistically significant. Continuous variables were described as the mean (standard deviation) or median (1st quartile–3rd quartile), and categorical variables were described as % (number). Differences in variables of interest between the created groups were evaluated by one-way analysis of variance and the Tukey–Kramer test or Kruskal–Wallis test and Steel–Dwass test for continues variables, and a chi-squared test for categorized variables. Effects of eating speed on the presence of low muscle strength, low skeletal muscle mass, and sarcopenia were evaluated using logistic regressions. Factors of age, sex, insulin usage, smoking, exercise, total energy intakes (kcal/IBW/day), history of cancer, history of heart diseases and CKD stage ≥4 were considered to be independent variables.

## 3. Results

In total, 563 individuals were enrolled in this study, though 324 were found to be ineligible; thus, this rendered a final number of 239 participants (Figure 1).

Clinical characteristics of the participants are shown in Table 1. Their mean age and BMI were 71.6 (6.3) years and 23.6 (3.5) kg/m^2^, respectively. According to the questionnaire, the frequency of fast, normal, and slow eating speeds was 47.3%, 32.2%, and 20.5%, respectively. From the body composition analyzer, the prevalence of low skeletal muscle mass, low muscle strength, and sarcopenia was 28.4%, 29.3%, and 15.9%, respectively.

Analyses showed that slow eaters had greater amounts of low skeletal muscle mass, low muscle strength, or sarcopenia than fast- or normal-speed eaters (Table 1 and Figure 2). No statistically significant differences in total energy or protein intakes were detected between the eating-speed groups, nor were there any significant effects of age or sex on the prevalence of low skeletal muscle mass, low muscle strength, or sarcopenia.

Relationships between eating speed and the presence of low skeletal muscle mass, low muscle strength, or sarcopenia are shown in Table 2. After adjusting for covariates, compared to slow eaters, the odds ratio of having low skeletal muscle mass among fast- and normal-speed eaters was 0.27 [95% confidence interval (CI): 0.12–0.57, *p* < 0.001] and 0.32 (95% CI: 0.14–0.71, *p* = 0.005), respectively; the odds ratio of having low muscle strength among fast- and normal-speed eaters was 0.28 (95% CI: 0.12–0.65, *p* = 0.003) and 0.22 (95% CI: 0.09–0.57, *p* = 0.002), respectively; and the odds ratio of having sarcopenia among the fast- and normal-speed eaters was 0.31 (95% CI: 0.12–0.80, *p* = 0.016) and 0.18 (95% CI: 0.06–0.53, *p* = 0.002), respectively (Model 4).

## 4. Discussion

Our study demonstrated, in line with the hypothesis, that a slow eating speed is associated with a heightened risk of low skeletal muscle mass, low muscle strength, and sarcopenia in older patients with T2D.

Earlier research has revealed that eating speed is strongly connected to body weight: fast eating is often tied to a high BMI and its associated comorbidities [18,23,26,27,28,29], whereas slow eating is related to a lean weight status [30]. Reasons for this phenomenon are not entirely clear, although it is feasible that fast eating engenders greater energy intake [31], whereas slow eating does the opposite [32]. Chewing food well over time is known to increase both the release of neuronal histamine and the postprandial response of hormones such as ghrelin, glucagon-like peptide-1, and peptide YY, which are involved in the stimulation of the satiety center and sympathetic nervous system [33], and the regulation of hunger, satiety, and energy intake, respectively [34]. In this study, we found that slow eaters had a lower BMI than those in the two other groups, yet dietary intake was comparable across all eating, whereas there were no relationship speed classifications. Discrepancies between our data and those of previous investigations could be due to reporting errors, as individuals with obesity tend to underestimate their dietary intakes [31,35]. Nevertheless, and more importantly here, our collective findings suggest that reduced intake is not implicated in the association between slow eating and sarcopenia; therefore, other factors are likely at play. To this point, several scholars have noticed a close relationship between sarcopenia and oral function [36,37,38,39,40], indicating that this muscle disorder might be linked to slower-paced eating via impairments in swallowing.

The limitations of this study should be mentioned. First, it was cross-sectional in nature; thus, a causal relationship remains to be elucidated. Second, data on participant eating speeds were collected using a self-report questionnaire, which is a subjective assessment and not an objective one, and therefore, our outcomes should be interpreted with caution. Third, the number of the participants was relatively small and outpatient-only. However, the proportion of the presence of sarcopenia of 15.8% in this study, was almost same as the data of a recent metanalysis [41]. Fourth, we did not have the data of frequency of hospital admissions, which are associated with muscle loss [42]. Lastly, considering that all participants had T2D, it is possible that having a slow eating speed could be attributed more to advice from healthcare professionals than sarcopenia itself.

## 5. Conclusions

In conclusion, slow eating was found to be associated with an increased risk of sarcopenia in older patients with T2D. Current recommendations for the prevention of conditions such as obesity [18], metabolic syndrome [26] and nonalcoholic fatty liver disease [23] in patients with T2D include expressed eating speed, and, as such, stand in opposition to our findings that slow eating could be harmful in some cases. Presently, no clear strategy exists for shifting nutritional guidance from obesity and/or metabolic syndrome to sarcopenia in older people with T2D [43]. Henceforth, prospective explorations into mechanisms underlying the linkage between slow eating and sarcopenia and the direction of causality will inform the optimization of the overall management of T2D in older patients going forward.

## Figures and Tables

**Figure 1 nutrients-14-00759-f001:**
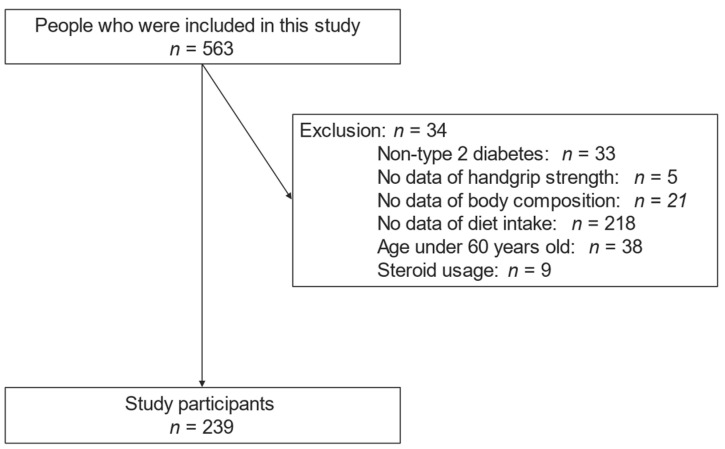
Inclusion and exclusion flow.

**Figure 2 nutrients-14-00759-f002:**
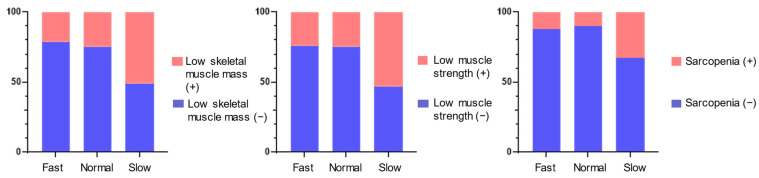
Proportions of low skeletal muscle mass, low muscle strength and sarcopenia among fast-, normal- and slow-speed eating groups.

**Table 1 nutrients-14-00759-t001:** Clinical characteristics of study participants.

	All *n* = 239	Fast, *n* = 113	Normal, *n* = 77	Slow, *n* = 49	*p*
Age, years	71.6 (6.2)	70.7 (6.2)	72.1 (6.2)	73.0 (6.0)	0.075
Men, % (*n*)	58.6% (140)	60.2% (68)	62.3% (48)	49.0% (24)	0.297
Duration of diabetes, years	18.4 (11.5)	17.9 (10.8)	18.8 (13.1)	19.1 (10.7)	0.776
Family history of diabetes, % (*n*)	41.8% (100)	46.0% (52)	28.6% (22)	53.1% (26)	0.012
Height, cm	160.7 (8.6)	161.4 (8.6)	161.3 (8.1)	158.0 (9.1)	0.048
Body weight, kg	61.1 (10.8)	63.3 (10.7)	61.5 (10.6)	55.2 (9.7) ^†^^‡^	<0.001
Body mass index, kg/m^2^	23.6 (3.5)	24.2 (3.1)	23.6 (3.8)	22.2 (3.8) ^†^^‡^	0.004
Appendicular muscle mass, kg	17.8 (4.0)	18.5 (4.1)	18.1 (3.8)	15.8 (3.7) ^†^^‡^	0.001
Skeletal muscle mass index, kg/m^2^	6.8 (1.0)	7.0 (1.0)	6.9 (1.1)	6.2 (0.9) ^†^^‡^	<0.001
Low skeletal muscle mass, % (*n*)	28.4% (68)	21.2% (24)	24.7% (19)	51.0% (25)	<0.001
Handgrip strength, kg	26.5 (8.3)	27.9 (8.7)	27.1 (7.2)	22.4 (7.7) ^†^^‡^	<0.001
Low muscle strength, % (*n*)	29.3% (70)	22.1% (25)	24.7% (19)	53.1% (26)	<0.001
Presence of sarcopenia, % (*n*)	15.9% (38)	12.4% (14)	10.4% (8)	32.7% (16)	0.001
Insulin, % (*n*)	23.5% (56)	23.2% (26)	23.4% (18)	24.5% (12)	0.984
GLP-1 antagonist, % (*n*)	8.4% (20)	12.5% (14)	5.2% (4)	4.1% (2)	0.097
SGLT2 inhibitor, % (*n*)	16.8% (40)	22.3% (25)	11.7% (9)	12.2% (6)	0.100
Smoker, % (*n*)	13.4% (32)	15.9% (18)	10.4% (8)	12.2% (6)	0.527
Exerciser, % (*n*)	50.6% (121)	51.3% (58)	52.0% (40)	46.9% (23)	0.843
History of cancer, % (*n*)	22.2% (53)	24.8% (28)	36.4 (28)	30.6 (15)	0.227
History of heart diseases, % (*n*)	29.7% (71)	21.2 (24)	23.4% (18)	22.5% (11)	0.940
CKD stage ≥4, % (*n*)	4.6% (11)	0.9% (1)	9.1% (7)	6.3% (3)	0.025
HbA1c, mmol/mol	54.3 (8.4)	54.9 (8.6)	53.8 (8.7)	53.9 (7.4)	0.683
HbA1c, %	7.1 (0.8)	7.2 (0.8)	7.1 (0.8)	7.1 (0.7)	0.683
Plasma glucose, mmol/L	8.1 (2.3)	8.0 (2.1)	8.1 (2.4)	8.3 (2.6)	0.678
Total energy intake, kcal/day	1778 (666)	1765 (649)	1849 (714)	1699 (627)	0.453
Total energy intake, kcal/kg IBW/day	31.3 (11.7)	30.7 (10.9)	32.6 (13.0)	30.8 (11.6)	0.523
Protein intake, g/day	75.9 (34.7)	74.8 (30.1)	81.4 (37.0)	69.8 (29.5)	0.130
Protein intake, % Energy	17.1 (3.5)	17.1 (3.5)	17.5 (3.6)	16.4 (3.1)	0.173
Fat intake, g/day	57.6 (25.3)	56.7 (24.1)	59.4 (26.2)	56.6 (27.1)	0.739
Fat intake, % Energy	29.2 (6.3)	29.0 (6.5)	29.1 (6.4)	29.6 (5.7)	0.862
Carbohydrate intake, g/day	221.7 (86.5)	223.3 (90.5)	229.6 (86.9)	205.5 (75.2)	0.303
Carbohydrate intake, % Energy	50.3 (8.7)	50.7 (9.1)	50.4 (8.2)	49.3 (8.6)	0.623
Alcohol consumption, g/day	0 (0–2.4)	0 (0–2.9)	0 (0–0.2)	0.1 (0–9.9) ^‡^	0.033
Dietary fiber intake, g/day	12.6 (5.4)	12.4 (5.4)	13.3 (5.6)	12.2 (5.2)	0.416

Data were expressed as mean (standard deviation), median (1st quartile―3rd quartile) or number (%). Differences in variables of interest between the created groups were evaluated by one-way analysis of variance and the Tukey–Kramer test or Kruskal–Wallis test and Steel–Dwass test for continues variables, and a chi-squared test for categorized variables. GLP-1, glucagon-like peptide-1; SGLT, sodium-glucose cotransporter; CKD, chronic kidney disease; IBW, ideal body weight. †, *p* < 0.05 vs. fast-speed eating and ‡, *p* < 0.05 vs. normal-speed eating by Tukey–Kramer test.

**Table 2 nutrients-14-00759-t002:** Relationship between eating speed and the presence of sarcopenia.

**The presence of low muscle mass**	**Model 1**		**Model 2**		**Model 3**		**Model 4**	
	**Odds ratio (95% CI)**	***p* value**	**Odds ratio (95% CI)**	***p* value**	**Odds ratio (95% CI)**	***p* value**	**Odds ratio (95% CI)**	***p* value**
Age (year)	―	―	1.06 (1.01–1.12)	0.011	1.07 (1.02–1.12)	0.009	1.07 (1.02–1.13)	0.009
Men	―	―	1.09 (0.60–2.01)	0.770	1.22 (0.65–2.28)	0.537	1.23 (0.65–2.32)	0.528
Insulin usage	―	―	―	―	1.43 (0.72–2.87)	0.303	1.49 (0.75–2.98)	0.259
Smoking	―	―	―	―	0.82 (0.31–2.14)	0.681	0.77 (0.29–2.03)	0.593
Exercise	―	―	―	―	1.66 (0.91–3.05)	0.106	1.64 (0.89–3.03)	0.116
Total energy intake (kcal/kg IBW/day)	―	―	―	―	0.97 (0.94–1.01)	0.092	0.97 (0.94–1.01)	0.089
CKD stage ≥4	―	―	―	―	―	―	0.68 (0.16–2.88)	0.600
History of cancer	―	―	―	―	―	―	0.73 (0.34–1.56)	0.415
History of heart diseases	―	―	―	―	―	―	0.84 (0.42–1.68)	0.625
Eating speed								
Fast	0.26 (0.13–0.53)	<0.001	0.28 (0.14–0.59)	<0.001	0.27 (0.13–0.56)	<0.001	0.27 (0.12–0.57)	<0.001
Normal	0.31 (0.15–0.67)	0.003	0.32 (0.15–0.69)	0.004	0.31 (0.14–0.68)	0.004	0.32 (0.14–0.71)	0.005
Slow	Reference	―	Reference	―	Reference	―	Reference	―
**The presence of low handgrip strength**	**Model 1**		**Model 2**		**Model 3**		**Model 4**	
	**Odds ratio (95% CI)**	***p* value**	**Odds ratio (95% CI)**	***p* value**	**Odds ratio (95% CI)**	***p* value**	**Odds ratio (95% CI)**	***p* value**
Age (year)	―	―	1.18 (1.12–1.25)	<0.001	1.19 (1.12–1.26)	<0.001	1.19 (1.12–1.27)	<0.001
Men	―	―	0.58 (0.30–1.11)	0.099	0.64 (0.33–1.25)	0.191	0.59 (0.29–1.18)	0.138
Insulin usage	―	―	―	―	1.85 (0.87–3.95)	0.111	1.88 (0.86–4.09)	0.112
Smoking	―	―	―	―	0.70 (0.22–2.19)	0.535	0.65 (0.20–2.15)	0.482
Exercise	―	―	―	―	1.58 (0.82–3.06)	0.175	1.70 (0.86–3.37)	0.129
Total energy intake (kcal/kg IBW/day)	―	―	―	―	1.00 (0.97–1.03)	0.761	1.00 (0.97–1.03)	0.923
CKD stage ≥4	―	―	―	―	―	―	4.37 (1.00–19.1)	0.005
History of cancer	―	―	―	―	―	―	0.37 (0.15–0.92)	0.033
History of heart diseases	―	―	―	―	―	―	1.40 (0.67–2.91)	0.372
Eating speed								
Fast	0.25 (0.12–0.51)	<0.001	0.28 (0.13–0.62)	0.002	0.27 (0.12–0.61)	0.002	0.28 (0.12–0.65)	0.003
Normal	0.29 (0.14–0.62)	0.002	0.28 (0.12–0.67)	0.004	0.27 (0.11–0.64)	0.003	0.22 (0.09–0.57)	0.002
Slow	Reference	―	Reference	―	Reference	―	Reference	―
**The presence of sarcopenia**	**Model 1**		**Model 2**		**Model 3**		**Model 4**	
	**Odds ratio (95% CI)**	***p* value**	**Odds ratio (95% CI)**	***p* value**	**Odds ratio (95% CI)**	***p* value**	**Odds ratio (95% CI)**	***p* value**
Age (year)	―	―	1.16 (1.09–1.24)	<0.001	1.18 (1.10–1.26)	<0.001	1.17 (1.09–1.26)	<0.001
Men	―	―	1.11 (0.51–2.43)	0.796	1.29 (0.57–2.90)	0.539	1.19 (0.52–2.73)	0.681
Insulin usage	―	―	―	―	1.53 (0.63–3.75)	0.351	1.63 (0.65–4.04)	0.295
Smoking	―	―	―	―	0.76 (0.19–2.97)	0.693	0.67 (0.17–2.72)	0.579
Exercise	―	―	―	―	2.35 (1.04–5.31)	0.040	2.54 (1.09–5.94)	0.032
Total energy intake (kcal/kg IBW/day)	―	―	―	―	0.99 (0.95–1.03)	0.564	0.99 (0.95–1.03)	0.700
CKD stage ≥4	―	―	―	―	―	―	2.33 (0.47–11.6)	0.299
History of cancer	―	―	―	―	―	―	0.36 (0.12–1.10)	0.072
History of heart diseases	―	―	―	―	―	―	1.29 (0.56–3.01)	0.550
Eating speed								
Fast	0.29 (0.13–0.66)	0.003	0.34 (0.14–0.81)	0.015	0.31 (0.12–0.76)	0.010	0.31 (0.12–0.80)	0.016
Normal	0.24 (0.09–0.62)	0.003	0.22 (0.08–0.60)	0.003	0.19 (0.07–0.55)	0.002	0.18 (0.06–0.53)	0.002
Slow	Reference	―	Reference	―	Reference	―	Reference	―

Model 1 was the unadjusted model, model 2 was adjusted for age and sex, model 3 was adjusted for model 2 and insulin usage, smoking, exercise and total energy intake, and model 4 was adjusted for model 3 and CKD stage ≥4, history of cancer and history of heart diseases. CKD, chronic kidney disease; IBW, ideal body weight.

## Data Availability

The data that support the findings of this study are available from the corresponding author, YH, upon reasonable request.

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
