# Peer review of "Eating Speed Is Associated with the Presence of Sarcopenia in Older Patients with Type 2 Diabetes: A Cross-Sectional Study of the KAMOGAWA-DM Cohort"

_nutrients, 2022, doi:10.3390/nu14040759_

Round 1
Reviewer 1 Report
The present study is designed To determine the relationship between eating speed and the presence of sarcopenia in elderly patients with type 2 diabetes.
It seems to me a very interesting idea although the results may not be very reliable. The main data on which the study is based is the speed of intake and it has not been quantified by objective methods but by subjective data from the patient himself.
The authors collected data using a standardized questionnaire was used to gather data on each patient's duration of T2D, and smoking and exercise habits. It would be interesting if the authors add this survey as supplementary material.
The data in Table 1 is somewhat confusing because in some parameters there seems to be no difference between fast and slow eaters. The authors summarize by saying: Analyzes showed that slow eaters had greater amounts of low skeletal muscle mass, low muscle strength, or sarcopenia than fast or normal speed eaters. different figures in normal eaters) Low skeletal muscle mass (24% vs 29% vs 25%), low muscle strength (28% vs 20% vs 28%), or sarcopenia (16% vs 9% vs 17%). I think it would be clearer if the comparisons were by pairs Figure 2 visually shows some differences that are not seen in Table 1. The percentages in Table 1 are U-shaped while in Figure 2 a linear decrease is observed I'm sorry, but I don't understand.
Table 2 should indicate the adjusted covariates in model 1, 2 and 3
Another limitation of the study is that it involves outpatients, so the prevalence of sarcopenia is low. I understand that they are patients with no other major pathology except diabetes. Possibly sarcopenia may be more related to the coexistence of other disabling diseases or chronic complications of diabetes or frequency of hospital admissions. All these variables have not been collected and I think that they are more important for the appearance of sarcopenia than the speed of ingestion. It would be interesting to add these factors if you have them.
Author Response
Response to Reviewer 1
It seems to me a very interesting idea although the results may not be very reliable. The main data on which the study is based is the speed of intake and it has not been quantified by objective methods but by subjective data from the patient himself.
Response
Thank you for your comment. As you say, the data of speed of intake has not been quantified by objective methods. Thus, we mentioned this point as one of the limitations of this study in the Discussion section as below.
Discussion
“Second, data on participants’ eating speeds were collected using a self-report questionnaire, which is a subjective assessment and not an objective one, and therefore, our outcomes should be interpreted with caution.”
The authors collected data using a standardized questionnaire was used to gather data on each patient's duration of T2D, and smoking and exercise habits. It would be interesting if the authors add this survey as supplementary material.
Response
Thank you for your comment. According to your comment, we have added the survey questionnaire as supplementary material.
The data in Table 1 is somewhat confusing because in some parameters there seems to be no difference between fast and slow eaters. The authors summarize by saying: Analyzes showed that slow eaters had greater amounts of low skeletal muscle mass, low muscle strength, or sarcopenia than fast or normal speed eaters. different figures in normal eaters) Low skeletal muscle mass (24% vs 29% vs 25%), low muscle strength (28% vs 20% vs 28%), or sarcopenia (16% vs 9% vs 17%). I think it would be clearer if the comparisons were by pairs Figure 2 visually shows some differences that are not seen in Table 1. The percentages in Table 1 are U-shaped while in Figure 2 a linear decrease is observed I'm sorry, but I don't understand.
Response
We are sorry for confusing with you. The categorized variables are show as number (%) in original manuscript. Slow eaters had greater amounts of low skeletal muscle mass (51.0% in slow vs. 21.2% in fast vs. 24.7% in normal), low muscle strength (53.1% in slow vs. 22.1% in fast vs. 24.7% in normal), or sarcopenia than fast or normal speed eaters (32.7% in slow vs. 12.4% in fast vs. 10.4% in normal) as shown Figure 2. We have revised categorized variables as % (number) to make it easier to understand.
Table 2 should indicate the adjusted covariates in model 1, 2 and 3
Response
Thank you for your suggestion. According to your suggestion, we have added indications of model 1, 2, 3 and 4 in Table2 described as below.
“Model 1 was unadjusted model, model 2 was adjusted age and sex, model 3 was adjusted for model2 and insulin usage, smoking, exercise and total energy intake, and model 4 was adjusted for model 3 and CKD stage ≥4, history of cancer and history of heart diseases.”
Another limitation of the study is that it involves outpatients, so the prevalence of sarcopenia is low. I understand that they are patients with no other major pathology except diabetes. Possibly sarcopenia may be more related to the coexistence of other disabling diseases or chronic complications of diabetes or frequency of hospital admissions. All these variables have not been collected and I think that they are more important for the appearance of sarcopenia than the speed of ingestion. It would be interesting to add these factors if you have them.
Response
Thank you for your suggestion. As you say, this study included the outpatients only. However, the proportions of the presence of sarcopenia of 15.8% in this study, which was almost same as the data of a recent metanalysis. Moreover, as you say, sarcopenia may be more related to the coexistence of other disabling diseases or chronic complications of diabetes or frequency of hospital admissions; thus, we have added the data of history of cancer, history of heart diseases and CKD stage ≥4, which were reported to be associated with the sarcopenia, whereas we did not have the data of frequency of hospital admissions. After adjusting for these data, the results were almost the same as the original one. We have added these points in the Methods, Results, including Table 2, and Discussion sections described as below.
Methods
“The estimated glomerular filtration rate (eGFR; mL/min/1.73 m2) was calculated using the Japanese Society of Nephrology equation [20] and eGFR under 30 mL/min/1.73 m2 were defined as chronic kidney diseases (CKD) stage ≥4 [21]. Past medical histories regarding heart diseases, including angina, coronary heart disease, heart failure, prior acute myocardial infarction, stroke (ischemic or hemorrhagic), and cancers, were obtained from electronic medical records.”
“Factors of age, sex, insulin usage, smoking, exercise, total energy intakes (kcal/IBW/day), history of cancer, history of heart diseases and CKD stage ≥4 were considered to be independent variables.”
Results
“After adjusting for covariates, compared to slow eaters, the odds ratio of having low skeletal muscle mass among fast and normal speed eaters was 0.27 [95% confidence interval (CI): 0.12–0.57, p < 0.001] and 0.32 (95% CI: 0.14–0.71, p = 0.005), respectively; the odds ratio of having low muscle strength among fast and normal speed eaters was 0.28 (95% CI: 0.12–0.65, p = 0.003) and 0.22 (95% CI: 0.09–0.57, p = 0.002), respectively; and the odds ratio of having sarcopenia among the fast and normal speed eaters was 0.31 (95% CI: 0.12–0.80, p = 0.016) and 0.18 (95% CI: 0.06–0.53, p = 0.002), respectively (Model 4).”
Discussion
“Third, the number of the participants were relatively small and outpatients only. However, the proportions of the presence of sarcopenia of 15.8% in this study, which was almost same as the data of a recent metanalysis [41]. Fourth, we did not have the data of frequency of hospital admissions, which are associated with muscle loss [42].”
References
- Matsuo, S., Imai, E., Horio, M., Yasuda, Y., Tomita, K., Nitta, K., et al. Revised equations for estimated GFR from serum creatinine in Japan. J. Kidney. Dis. 53, 982–992 (2009)
- Chen, T.K., Knicely, D.H., & Grams, M.E. Chronic Kidney Disease Diagnosis and Management: A Review. JAMA. 322, 1294–1304 (2019)
- Chung, S. M., Moon, J. S., & Chang, M. C. Prevalence of Sarcopenia and Its Association With Diabetes: A Meta-Analysis of Community-Dwelling Asian Population. Frontiers in medicine. 8, 681232 (2021)
- Kimura, T., Okamura, T., Iwai, K., Hashimoto, Y., Senmaru, T., Ushigome, E., et al. Japanese radio calisthenics prevents the reduction of skeletal muscle mass volume in people with type 2 diabetes. Open. Diabetes. Res. Care. 8, e001027 (2020)

Reviewer 2 Report
This is a very great study, which I have a minor revisions.
Introduction and discussion is ver well written. However, the methods must be better explained. For example, To add the analysis of association used in the statistical analyses.
In addition, in the limitation section, the authors need to clarify that eating speed questionnaire is a limitation, due to fact of use (self-reported).
Author Response
Response to Reviewer 2
Introduction and discussion is very well written. However, the methods must be better explained. For example, To add the analysis of association used in the statistical analyses.
Response
Thank you for your suggestion. According to your suggestion, we have revised the Methods section described as below.
Methods
“Differences in variables of interest between the created groups were evaluated by one-way analysis of variance and the Tukey–Kramer test or Kruskal–Wallis test and Steel–Dwass test for continues variables, and a chi-squared test for categorized variables.”
In addition, in the limitation section, the authors need to clarify that eating speed questionnaire is a limitation, due to fact of use (self-reported).
Response
Thank you for your suggestion. As you say, eating speed questionnaire is a limitation, due to self-reported data. Thus, we mentioned this point as a one of the limitations of this study in the Discussion section as below.
Discussion
“Second, data on participants’ eating speeds were collected using a self-report questionnaire, which is a subjective assessment and not an objective one, and therefore, our outcomes should be interpreted with caution.”